# Direct Burr Hole Access for Transverse–Sigmoid Junction DAVF Embolization: A Case Report

**DOI:** 10.3390/brainsci13060871

**Published:** 2023-05-27

**Authors:** James Withers, Robert W. Regenhardt, Adam A. Dmytriw, Justin E. Vranic, Rudolph Marciano, Christopher J. Stapleton, Aman B. Patel

**Affiliations:** 1College of Osteopathic Medicine, University of New England, Biddeford, ME 04005, USA; 2Department of Neurosurgery, Massachusetts General Hospital, Harvard Medical School, Boston, MA 02115, USA; robert.regenhardt@mgh.harvard.edu (R.W.R.); admytriw@mgh.harvard.edu (A.A.D.); jvranic@mgh.harvard.edu (J.E.V.); cstapleton@mgh.harvard.edu (C.J.S.); abpatel@mgh.harvard.edu (A.B.P.); 3Department of Neurology, Massachusetts General Hospital, Harvard Medical School, Boston, MA 02115, USA; 4Department of Radiology, Massachusetts General Hospital, Harvard Medical School, Boston, MA 02115, USA; 5Northern Light Neurosurgery and Spine, Bangor, ME 04401, USA; rmarcianoiii@northernlight.org

**Keywords:** DAVF, dural arteriovenous fistula, neurosurgery, burr hole, endovascular surgery

## Abstract

Dural arteriovenous fistulas (DAVFs) are rare intracranial vascular malformations that present with a variety of clinical signs and symptoms. Among these, intracranial hemorrhage is a severe complication. A 72-year-old male presented with headache and pulsatile tinnitus. Cerebral angiography revealed a Borden II/Cognard IIa+b DAVF. He underwent stage 1 transarterial embolization of the occipital artery which reduced shunting by 30%. Several attempts were made to access the fistula during stage 2 transvenous embolization, but it was not possible to access the left transverse sinus fistula site since there was no communication across the torcula from the right transverse sinus and the left inferior sigmoid–jugular bulb was occluded. Therefore, a single burr hole was drilled and direct access to the DAVF was achieved with a micropuncture needle under neuronavigational guidance. The left transverse–sigmoid sinus junction was then embolized with coils. After the procedure, angiography revealed that the DAVF was cured with no residual shunting. This case demonstrates how minimally invasive surgery provides an alternative method to access a DVAF when conventional transarterial and/or transvenous embolization treatment options are not possible. Each DAVF case has unique anatomy and physiology, and creative multi-disciplinary strategies can often yield the best results.

## 1. Introduction

Dural arteriovenous fistulas (DAVFs) are uncommon cerebral vascular lesions within the cranial cavity characterized by abnormal arteriovenous shunting between dural arteries and veins. DAVFs are acquired lesions and can present with a variety of clinical signs and symptoms [1,2,3]. The most feared outcomes are intracranial hemorrhage and death [2]. A retrospective study indicated that smoking was common in patients with DAVFs that progressed to intracranial hemorrhage [2]. In their study, males were more likely to present with intracranial hemorrhage while the prevalence of DAVFs was higher in females [1,2]. Understanding the anatomy of DAVFs is crucial to predicting the natural history and making treatment decisions [1].

Intracranial venous drainage can be disrupted depending on the location of the fistula. A meta-analysis of 377 patients found that DAVFs associated with aggressive neurological behaviors were more likely located at the tentorial incisure; aggressive presentations were most often related to hemorrhagic or non-hemorrhagic stroke [1,4]. This same study demonstrated that DAVFs least likely to present with aggressive neurological behaviors were located at the transverse–sigmoid sinuses and cavernous sinuses [5]. The anatomical location may also be associated with cortical venous reflux [1]. Furthermore, the location influences the presentation of the DAVF, including pulsatile tinnitus, bruits, headache, visual changes, cranial nerve palsies, motor or sensory deficits, mental status changes, seizures, and myelopathy, with tinnitus and headache being the most common [6,7]. However, not all patients are symptomatic [7]. 

Potential risk factors for DAVFs include venous thrombosis, thrombophlebitis, infectious mastoiditis, trauma, neoplasms, and cranial surgery [7]. Intracranial hemorrhage is one of the most serious clinical manifestations of DAVFs and can lead to severe neurological deficits and death [2,3]. The incidence of hemorrhage in patients with DAVFs is 7–33% [2,4,6]. The prognosis for patients presenting with intracranial hemorrhage is poor, with mortality ranging from 20% to 30% [1]. Management options for DAVFs range from observing symptoms in asymptomatic benign fistulas to obliteration via open surgery, endovascular embolization, and/or radiosurgery for more aggressive lesions [1,3]. Patients with mild clinical symptoms and low-flow fistulas that do not result in cortical venous reflux may be managed conservatively [3]. The decision to treat a DAVF is based on clinical presentation, lesion location, and the natural history of the fistula [3]. Patients who present with neurological deficits or have high-risk features, such as cortical venous reflux, are recommended to undergo intervention [3].

Due to the scarcity of long-term data on these relatively rare lesions, determining the treatment strategy typically relies on the severity of neurological symptoms and angiographic features [2]. The Borden and Cognard classifications are used to grade the severity of the shunting and the clinical risks of DAVFs. Benign DAVFs, such as Borden I/Cognard I and IIa DAVFs may only require observation [1,6]. DAVFs with no cortical venous reflux and with a benign presentation can be conservatively managed [1]. However, higher-grade DAVFs, such as Borden II and III/Cognard IIb–V, should undergo treatment [1,6]. Although transarterial and/or transvenous embolization is often the preferred initial management option for DAVFs, such treatment options are not always possible [8]. Direct surgical access is an alternative when percutaneous access is not feasible. The case described in this report employed the use of a single burr hole for direct cannulation of the left transverse sinus for the treatment of a DAVF when percutaneous transfemoral venous access was not possible [4,9]. The recommended goal of treatment should be to completely close the shunt regardless of the modality [7]. 

## 2. Case Report

A 72-year-old male presented with the primary complaint of persistent headache for two months. Additionally, he complained of gradually progressing “brain fog” and fatigue over the course of several years. There was also left-sided pulsatile tinnitus that subtly existed over several years but worsened in the preceding two months. CT and CTA head and neck were obtained, which showed possible early filling of the right cavernous sinus, increased vascularity around the left sigmoid–transverse sinus junction, and dilated bilateral cortical veins, more notable on the left (Figure 1). The patient was recommended to undergo cerebral angiography which revealed a Borden II/Cognard IIa+b DAVF (Figure 2).

Arterial supply was from the occipital artery, the middle meningeal artery, the ascending pharyngeal artery, the inferolateral trunk, and the artery of the falx cerebelli. Venous drainage was observed to flow retrograde in the sinus system, and there was substantial cortical venous reflux. Given the risk of intracranial hemorrhage and the presence of cortical venous reflux, the patient was recommended to undergo endovascular intervention for treatment. Informed consent was obtained after carefully discussing both the benefits and risks of this approach. The patient’s symptoms were managed with analgesics and his blood pressure was closely monitored along with his neurological status. 

The patient underwent stage one transarterial embolization under general anesthesia. Given the relative safety of occipital artery embolization, this approach was selected for the first attempt despite acknowledging that transvenous embolization may ultimately be necessary if a cure was not possible by a transarterial approach. An Apollo microcatheter with a 15 mm detachable tip was used for Onyx-34 embolization of the left occipital artery. Embolization was continued until the maximum reflux limit was reached. Systolic blood pressure was maintained at <140 mmHg post procedure. After stage 1 treatment, there was a 30% decrease in arteriovenous shunting, which resulted in improved tinnitus and headache. However, there was persistent fatigue and anxiety.

Given the incomplete occlusion of the DAVF in stage one, the patient was scheduled to undergo a second stage treatment with venous embolization of the fistula four weeks after the initial intervention. At that time, angiography confirmed persistent residual Borden II/Cognard IIa+b DAVF supplied predominantly by additional left occipital artery branches, the left inferolateral trunk, and the left posterior meningeal artery. Numerous dilated and tortuous cortical veins with delayed contrast washout were present, consistent with underlying global venous hypertension. Furthermore, there was no filling of the left transverse sinus during the venous phase, confirming it was not being utilized for normal venous egress and that it was an appropriate target for transvenous embolization. Several attempts were made to access the left transverse sinus. There was no communication across the torcula from the contralateral right transverse sinus (Figure 2C). Furthermore, the left inferior sigmoid sinus was occluded without a connection to the ipsilateral internal jugular vein (Figure 2D). Therefore, transfemoral percutaneous catheterization of the left transverse and sigmoid sinus for embolization was not possible. 

The following day, direct access to the left transverse sinus was obtained with neurosurgical burr hole access. In the operating room, the patient was placed in the supine position with his head turned to the right. His hair was shaved, lines were drawn for the incision location, and the patient was covered with sterile draping. Under general anesthesia, an initial incision was made to the bone to gain access to the skull. A high-speed drill was used to create the burr hole over the left transverse sinus. Under Brain Lab neuronavigational guidance, the left transverse sinus was located and a micropuncture needle was used to access the sinus. A microwire was carefully advanced under fluoroscopy through the needle into the target segment of the transverse sinus (Figure 3). Once placed, the needle was removed, and a micropuncture sheath was inserted over the wire. A microcatheter was advanced over a microwire through the micropuncture sheath into the left transverse sinus. Under roadmap guidance, coils were placed with no evidence of residual arteriovenous shunting at the end of the procedure (Figure 3).

The burr hole was covered, the tissue was closed, and antibiotics were applied to the wound. Systolic blood pressure was maintained at <140 mmHg post procedure. The patient was then transferred to the ICU for monitoring following the surgical intervention. Following the stage two definitive embolization of the fistula through the burr hole access, there was no evidence of complications post procedure. After two days of monitoring in the hospital, the patient was discharged home. 

During his two-week follow-up appointment, he presented with delirium and a “change in his personality”. Of note, imaging did not show evidence of hemorrhage or infarction. Given the concern for a possible relationship to thrombosis associated with occlusion of the high-flow fistula, he was treated with Lovenox and then transitioned to Eliquis. His delirium gradually resolved. The patient and his spouse reported significant improvement in his memory one year following stage two embolization. However, he still suffered from occasional mild headaches and fatigue, but he has overall improved from his initial presentation.

## 3. Discussion

DAVFs are relatively rare intracranial vascular lesions, accounting for 7–15% of all intracranial malformations, with a variety of clinical signs and symptoms. Abnormalities can be identified in many regions along the dura of the brain [4,10]. However, they are often found in the transverse, sigmoid, and cavernous sinuses. Although symptoms can be benign in some cases, DAVFs can lead to intracranial hemorrhage, which is one of the most serious clinical manifestations [4]. Various types of hemorrhage can occur from DAVFs as they traverse different compartments, including subdural, subarachnoid, and intraparenchymal hemorrhage [4]. A previous study assessed the potential clinical and angio-architectural risk factors of intracranial hemorrhage in patients with DAVFs. Demographic factors, such as age and sex, were related to hemorrhage risk, with male patients older than 50 years at increased risk. Clinical risk factors for DAVF-related intracranial hemorrhage included diabetes, alcohol, smoking, hypertension, and hyperlipidemia. However, after adjustment, the venous drainage pattern was the only independent risk factor for hemorrhage occurrence [3]. 

Arterial supply to DAVFs usually originates from the meningeal arteries, whereas their venous drainage occurs directly into the dural venous sinuses or through the cortical and meningeal veins [4]. CT angiography (CTA) and magnetic resonance angiography (MRA) can facilitate the screening and localization of DAVFs, aiding in their classification and risk stratification for hemorrhage. Additionally, these imaging modalities can be useful in assessing treatment response. CTA is readily accessible and can be employed within 24 h of patient presentation at most institutions [4,6,11]. However, digital subtraction angiography (DSA) remains the gold standard and should be pursued if there is a high index of suspicion for DAVF. DSA for the patient discussed in our case showed arterial supply to the DAVF from occipital artery branches, the left inferolateral trunk, and the artery of the falx cerebelli. There was retrograde flow in the venous sinus system with cortical venous reflux. As discussed, DAVFs with retrograde drainage in the cerebral cortical veins, cerebellar cortical veins, or both represent a serious medical condition that may lead to intracranial hypertension with an elevated risk of intracranial hemorrhage [4].

The Borden classification system groups DAVFs into three types based on the location of venous drainage and the presence or absence of cortical venous reflux. Borden type I DAVFs drain directly into a major venous sinus or meningeal vein with no cortical venous reflux. Type I DAVFs are generally considered benign, although a recent study demonstrated that a minority of patients may present with aggressive symptoms [10]. Borden type II DAVFs drain retrograde into the dural venous sinuses and/or cortical veins with associated cortical venous reflux due to high pressure. Borden type 3 lesions cause significant cortical venous reflux by draining directly into the cortical veins [1,2,6]. The Cognard system groups DAVFs into five types based on the direction of venous sinus drainage, the presence or absence of cortical venous reflux, and the type of venous outflow. Cognard type I DAVFs have anterograde flow with no cortical venous reflux. Cognard type IIa DAVFs drain into a sinus with retrograde flow into the sinus. Cognard type IIb DAVFs have retrograde flow into a cortical vein. Cognard type IIa+b DAVFs are characterized by retrograde flow into the sinus and cortical veins. Cognard type III DAVFs drain into a non-ectatic cortical vein. Cognard type IV DAVFs drain into a spinal perimedullary vein [1,6]. 

Endovascular embolization is the preferred method of treatment for most patients with DAVFs [10]. Endovascular treatment includes transarterial embolization, transvenous embolization, or a combination of the two methods [3,7]. General anesthesia is usually utilized for embolization to minimize patient movement and for pain control when certain embolic agents are utilized [3]. Endovascular treatment of some DAVFs may involve occluding a sinus using a deconstructive approach from percutaneous transvenous access. This is typically reserved for DAVFs with multiple arterial feeders, especially when some feeders represent a high risk for transarterial embolization and when the targeted sinus is not being used for normal brain drainage due to the pathology. Established techniques include the use of coils, especially for sinus take down, or Onyx, especially for arterial feeder occlusion [8].

Surgery may be recommended in patients with anterior fossa DAVFs or with complex lesions [3]. However, there is literature supporting endovascular approaches in the anterior fossa as well [6]. There are various surgical treatment options for DAVFs, including a single-stage craniotomy to expose the lesion for direct access. An approach often used is surgical clip placement on the draining vein. Previous studies have demonstrated that an adequate craniotomy can also be performed to control massive bleeding and obliterate feeding vessels [3]. Another study reported the use of surgically assisted transvenous embolization [9]. Similar to our case, this approach utilized surgical techniques after multiple unsuccessful endovascular attempts to traverse the occluded left sigmoid sinus [9]. There were, however, several differences compared to our report, including the lesion anatomy and our use of a single burr hole instead of a craniotomy for direct access [9].

Our patient presented with a Borden II/Cognard IIa+b DAVF which is classified as aggressive with cortical venous reflux and retrograde flow involving the dural sinus [6]. The goal of treatment is to occlude arteriovenous shunting by obliteration of the fistula. Conventional treatment of DAVFs can be accomplished with transarterial and/or transvenous embolization. However, sufficient access to the fistula site is required for treatment [7]. The patient was initially treated with transarterial Onyx embolization of the occipital artery. Cognard and colleagues describe Onyx embolization to be an appropriate treatment of choice for DAVFs [6]. Previous studies have also demonstrated that surgical approaches serve as a safe and effective alternative to embolization [12]. Surgical considerations become especially important when access to DAVFs may be limited by the presence of restrictive or occlusive vasculature en route to the fistula, such as thrombosis [9]. Our presented case highlights a unique multi-disciplinary approach in which a minimally invasive burr hole allowed microcatheter placement at the site of the fistula. 

## 4. Conclusions

The case report highlights the use of a burr hole approach for direct cannulation to access the site of a DAVF, which can be considered when traditional embolization methods are not possible due to restrictive or occlusive vasculature. The goal of DAVF treatment is to occlude arteriovenous shunting by obliteration of the fistula. Traditional methods of treatment include transarterial embolization, transvenous embolization, and/or open surgery. The choice of treatment depends on the location, angioarchitecture, and physiology of the DAVF, as well as the patient’s overall health and preferences [13,14]. Our case highlights that a burr hole approach for direct access may be safe, effective, and minimally invasive, providing a potential alternative treatment option [3,4,9]. Furthermore, this case report emphasizes the importance of a multi-disciplinary approach when treating patients with DAVFs, as each case presents unique anatomy and physiology. Creative strategies involving a team of specialists, including neurosurgeons, radiologists, and neurologists, can often yield the best results in managing these rare and complex vascular malformations.

## Figures and Tables

**Figure 1 brainsci-13-00871-f001:**
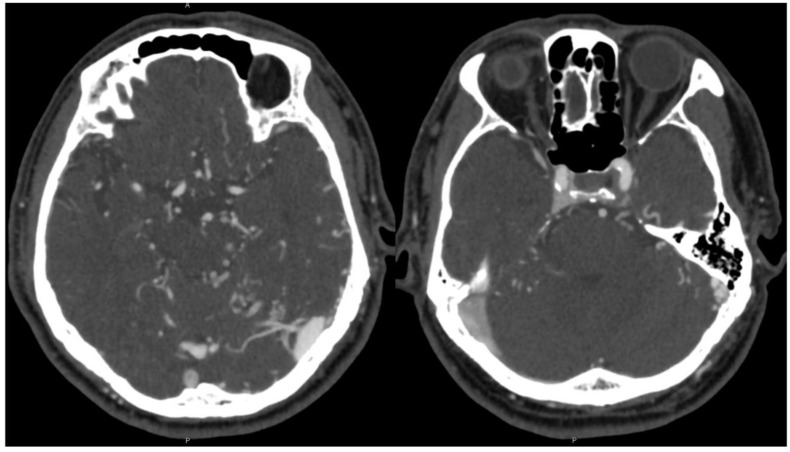
CTA demonstrated early filling of the right cavernous sinus, increased vascularity around the left sigmoid–transverse sinus junction, and dilated bilateral cortical veins, more notable on the left.

**Figure 2 brainsci-13-00871-f002:**
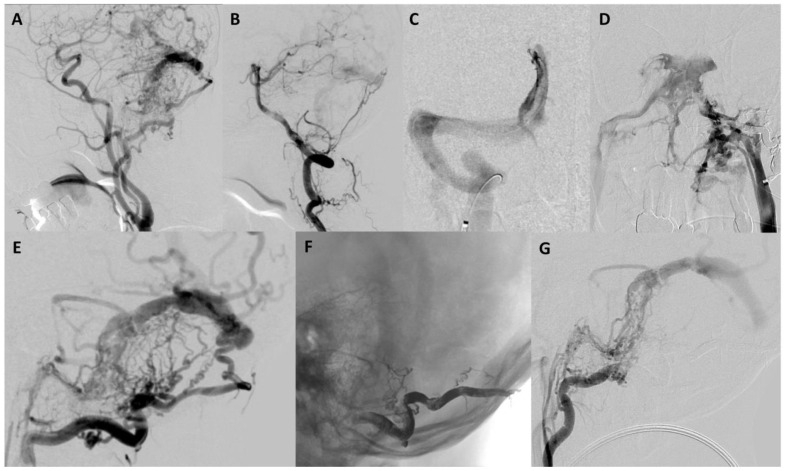
(**A**) Lateral left common carotid angiography showed a left transverse–sigmoid junction dural arteriovenous fistula (Borden II/Cognard IIa+b) supplied by multiple occipital artery branches, the middle meningeal artery, the ascending pharyngeal artery, and the inferolateral trunk. (**B**) Lateral left vertebral angiography showed there is also supply from the artery of the falx cerebelli. (**C**) Anteroposterior torcula venography showed there is no communication from the right transverse sinus to the left transverse sinus. (**D**) Anteroposterior left internal jugular venography showed there is no communication with the left sigmoid sinus consistent with occlusion. (**E**) Lateral left occipital angiography showed multiple feeders to the left transverse–sigmoid junction dural arteriovenous fistula. (**F**) The lateral unsubtracted image showed the Onyx cast in the occipital artery after embolization. (**G**) Lateral left occipital angiography post Onyx embolization showed a reduction in arteriovenous shunting by about 30%.

**Figure 3 brainsci-13-00871-f003:**
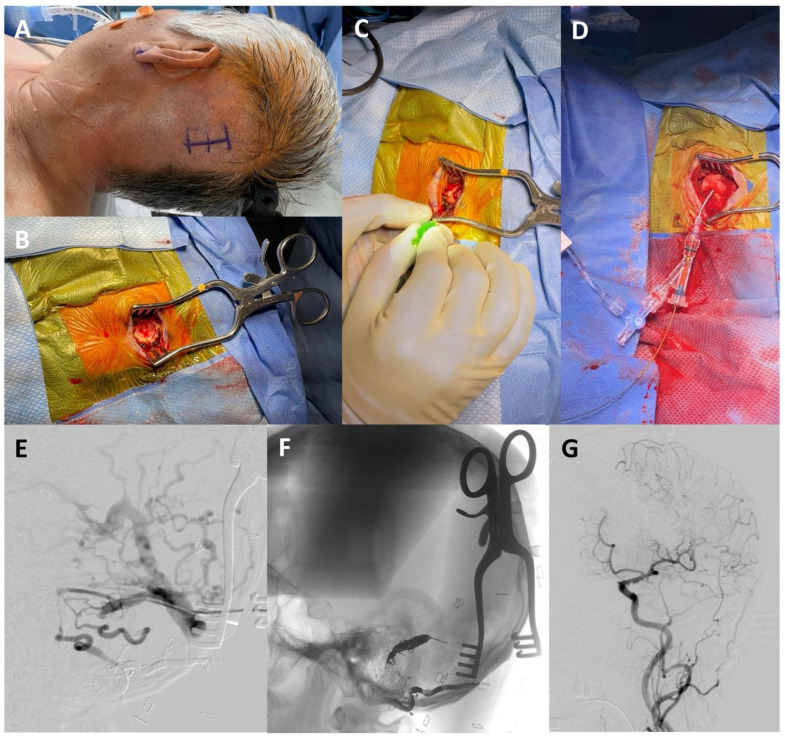
(**A**) Head positioning for the burr hole procedure. (**B**) A small incision was made, and a burr hole was drilled. (**C**) Direct sinus access was achieved with a micropuncture needle. (**D**) Catheterization of the left sigmoid sinus. (**E**) Lateral oblique left sigmoid venography showed good catheter placement in the left sigmoid sinus. (**F**) The lateral oblique unsubtracted image showed the coil mass after embolization of the left transverse–sigmoid junction. (**G**) Final lateral oblique left common carotid angiography showed cure of the dural arteriovenous fistula with no residual shunting.

## Data Availability

Any additional data will be made available upon reasonable request to the corresponding author pending IRB approval.

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
