# Peer review of "Direct Burr Hole Access for Transverse–Sigmoid Junction DAVF Embolization: A Case Report"

_brainsci, 2023, doi:10.3390/brainsci13060871_

Round 1

Reviewer 1 Report

THE authors present a case report on a DAVFs at the sigmoid-jugular junction. As uncommon pathologies from the point of view of treatment, I always find these reports very useful. However, the paper has numerous shortcomings:

First of all, the authors should better clarify from the introduction the epidemiology of the sites where DAVFs occur and their clinical onset, not all type 2s are treated, and the management and choice of treatment should be better explained. I doubt that the patient had angiography as the first diagnostic test, I think the clinical onset should be better explained in the case description, whether the patient had speech or psychiatric or cognitive disorders given the presence of frequent blood theft for this type of pathology (Armocida D, Palmieri M, Paglia F, Berra LV, D'Angelo L, Frati A, Santoro A. Rapidly progressive dementia and Parkinsonism as the first symptoms of dural arteriovenous fistula. The Sapienza University experience and comprehensive literature review concerning the clinical course of 102 patients. Clin Neurol Neurosurg. 2021 Sep;208:106835. doi: 10.1016/j.clineuro.2021.106835. Epub 2021 Jul 22. PMID: 34364030.) Has the patient had CT, angio-CT, or PET-CT? Insert images. How quickly did the patient recover from symptoms?

The case needs to be better discussed from these perspectives.

Author Response

Thank you very much for the review and the suggested edits. We incorporated your suggestions into the manuscript. The following are the changes made:

  1. Epidemiology was clarified in the introduction and the conclusion, including percent of total vascular malformations. Risk factors for complications were also added, such as smoking.
  2. The fourth paragraph of the introduction now states that type I and IIa “DAVFs may only require observation. DAVFs with no cortical venous reflux and with a benign presentation can be conservatively managed.”
  3. Initial clinical presentation of the patient was expanded to include the timing of his symptoms as well as other associated symptoms. We have referenced the excellent suggested manuscript (Armocida D, Palmieri M, Paglia F, Berra LV, D'Angelo L, Frati A, Santoro A. Rapidly progressive dementia and Parkinsonism as the first symptoms of dural arteriovenous fistula. The Sapienza University experience and comprehensive literature review concerning the clinical course of 102 patients. Clin Neurol Neurosurg. 2021 Sep;208:106835. doi: 10.1016/j.clineuro.2021.106835. Epub 2021 Jul 22. PMID: 34364030). The description of the diagnostic tools utilized was also expanded and included a CTA. We now include a new figure (now called Figure 1) that shows the CTA.
  4. We now added details about the patient’s recovery, including the timing of symptoms improvement.
  5. Information regarding treatment options was added to the introduction.

Reviewer 2 Report

Review to the case report Direct Burr Hole Access for Transverse-Sigmoid

Junction DAVF Embolization: A Case Report

I reviewed this case report with great interest. Withers et al. present an interesting case report about treatment of a 72-year-old male with an dural arteriovenous fistulas (DAVFs) typ Borden II/Cognard IIa+b.

DAVFs are rarely vascular malformations. Nevertheless, they could lead to disastrous intracranial hemorrhage. Typically, DAVFs were treated surgically or interventionally or in a combined procedure. The authors present the use of a burr hole approach for direct cannulation to access the site of a DAVF. The case report is structured and well-written. All procedures are present in a adequately manner and the used pictures are selected in a good manner.

I had only a few queries:

Did the authors had information about long-term outcome of the patient (> 6 months after treatment)? They should add this statement to the case report.

What treatment of systolic blood pressure did the intensivist choose to avoid complications after final treatment (I think about “walk-through” phenomena)?

The used pictures are selected in a good manner, however the authors should also present a MRI image to a optimize visualization of the localization of the DAVF.

Author Response

Thank you very much for the review and the suggested edits. We incorporated your suggestions into the manuscript. The following are the changes made:

  1. The long-term outcome of the patient was added to the case report.
  2. After embolization systolic blood pressure was maintained <140 mmHg. These details were added to the case report.
  3. During the work-up of this patient, an MRI was not performed. We do, however, now include a description of the initial CTA and include it in a new figure, labeled Figure 1.

Reviewer 3 Report

Thank you very much for the opportunity to review the publication, “Direct Burr Hole Access for Transverse-Sigmoid Junction DAVF Embolization: A Case Report.” The authors describe a unique case highlighting the multimodal treatment of a patient with a complex transverse sigmoid dAVF.  

Introduction

It would be valuable to cite the CONDOR data. This is by far the most robust data on these lesions. 

Case Report

Would be valuable to discuss the stage 1 procedure more. Type of onyx, was a balloon used, was venous access attempted at this time?

There is a clear communication from the left transverse to sigmoid to jugular vein. It looks tenuous but it is there. That is different from it being not navigable. 

An arterial injection but venous phase would be valuable to demonstrate that there is no use of the left TS sinus, ie how do those cortical veins like labbe drain? If there is no use of the normal brain seems like upfront venous sacrifice would have been the best option

Discussion

No particular issues

Figure 1

Small point – panel B, this looks like the artery of the falx cerebelli. Would need to see a frontal view but it looks like this is on the falx cerebelli and not posterior dura.

There is also supply from the MMA (panel A) as well as the ascending pharyngeal artery (Panel G). 

Author Response

Thank you very much for the review and the suggested edits. We incorporated your suggestions into the manuscript. The following are the changes made:

  1. The CONDOR study was added and is now sited.
  2. We have now added details of the stage 1 procedure: “The patient underwent stage 1 transarterial embolization under general anesthesia. Given the safety of occipital artery embolization, this approach was selected for the first attempt despite acknowledging that transvenous embolization may ultimately be necessary if cure was not possible a transarterial approach. An Apollo microcatheter with a 15mm detachable tip was used for Onyx-34 embolization of the left occipital artery. Embolization was continued until the maximum reflux limit was reached. After stage 1 treatment, there was a 30% decrease in arteriovenous shunting…”
  3. New Figure 2, Panel D shows there is no communication from the left jugular vein to the left sigmoid sinus. Furthermore, New Figure 3, Panel E shows that there is no drainage from the left transverse-sigmoid junction (after direct cannulation) into the left jugular.
  4. We now state there was no filling of the left transverse sinus during the venous phase, confirming it was not being utilized for normal venous egress and that it was an appropriate target for transvenous embolization.
  5. Thank you for these important clarifications for the Figure. We have made the proposed changes to the figure legend.